# mRNA Vaccines Enhance Neutralizing Immunity against SARS-CoV-2 Variants in Convalescent and ChAdOx1-Primed Subjects

**DOI:** 10.3390/vaccines9080918

**Published:** 2021-08-18

**Authors:** Dorit Fabricius, Carolin Ludwig, Judith Scholz, Immanuel Rode, Chrysanthi Tsamadou, Eva-Maria Jacobsen, Martina Winkelmann, Aline Grempels, Ramin Lotfi, Aleš Janda, Sixten Körper, Guido Adler, Klaus-Michael Debatin, Hubert Schrezenmeier, Bernd Jahrsdörfer

**Affiliations:** Division of Immune Cell Therapeutics, Institute of Clinical Transfusion Medicine and Immunogenetics, Helmholtzstrasse 10, 89081 Ulm, Germany; dorit.fabricius@uni-ulm.de (D.F.); c.ludwig@blutspende.de (C.L.); judith.scholz@uni-ulm.de (J.S.); i.rode@blutspende.de (I.R.); c.tsamadou@blutspende.de (C.T.); eva-maria.jacobsen@uni-ulm.de (E.-M.J.); m.winkelmann@blutspend.de (M.W.); a.grempels@blutspende.de (A.G.); r.lotfi@blutspende.de (R.L.); ales.janda@uniklinik-ulm.de (A.J.); s.koerper@blutspende.de (S.K.); guidoadler46@gmail.com (G.A.); klaus-Michael.Debatin@uniklinik-ulm.de (K.-M.D.); h.schrezenmeier@blutspende.de (H.S.)

**Keywords:** heterologous vaccination regimes using mRNA vaccines may allow enhanced protection against SARS-CoV-2, including current VOCs, RNA vaccines may facilitate rapid (re-) qualification of convalescent plasma donors with high titers of broadly neutralizing antibodies

## Abstract

To identify the most efficient methods of immunological protection against SARS-CoV-2, including the currently most widespread variants of concern (VOCs)—B.1.1.7, B.1.351 and P.1—a simultaneous side-by-side-comparison of available vaccination regimes is required. In this observational cohort study, we compared immunological responses in 144 individuals vaccinated with the mRNA vaccines BNT162b2 or mRNA-1273 and the vector vaccine ChAdOx1-nCoV-19, either alone, in combination, or in the context of COVID-19-convalescence. Unvaccinated COVID-19-convalescent subjects served as a reference. We found that cellular and serological immune responses, including neutralizing capacity against VOCs, were significantly stronger with mRNA vaccines as compared with COVID-19-convalescent individuals or vaccinated individuals receiving the vector vaccine ChAdOx1-nCoV-19. Booster immunizations with mRNA vaccines triggered strong and broadly neutralizing antibody and IFN-γ responses in 100% of vaccinated individuals investigated. This effect was particularly strong in COVID-19-convalescent and ChAdOx1-nCoV-19-primed individuals, who were characterized by comparably moderate cellular and neutralizing antibody responses before mRNA vaccine booster. Heterologous vaccination regimes and convalescent booster regimes using mRNA vaccines may allow enhanced protection against SARS-CoV-2, including current VOCs. Furthermore, such regimes may facilitate rapid (re-)qualification of convalescent plasma donors with high titers of broadly neutralizing antibodies.

## 1. Introduction

The current COVID-19 pandemic has launched a worldwide intensive effort to develop efficient vaccines, which turned out to be a particular challenge in the face of emerging novel SARS-CoV-2 variants of concern (VOCs). Meanwhile, the European Centre for Disease Prevention and Control lists four VOCs on its website, which have begun to supersede the original SARS-CoV-2 wildtype: B.1.1.7 (Alpha, first detected in the United Kingdom), B.1.351 (Beta, first detected in South Africa), P.1 (Gamma, first detected in Brazil), and B.1.617.2 (Delta, first detected in India) (https://www.ecdc.europa.eu/en/covid-19/variants-concern (accessed on 1 July 2021)).

By the end of February 2021, when this study was initiated, three COVID-19 vaccines had been approved in Germany, and are currently administered worldwide. Two mRNA vaccines from Pfizer/BioNTech and Moderna are currently available—the former approved in Europe on 21 December 2020, the latter on 6 January 2021—with reported efficacy results in phase 2/3 trials of 95% (Pfizer/BioNTech) [1] and 94.1% (mRNA-1273) [2]. The mRNA vaccine BNT162b2 by BioNTech (trade name Comirnaty) is approved in Europe for individuals aged ≥ 12 years, whereas the vaccine mRNA-1273 by Moderna (trade name COVID-19 Vaccine Moderna) is approved for individuals aged ≥ 18 years. Both vaccines contain mRNA encoding the SARS-CoV-2 spike protein, which is embedded in lipid nanoparticles and which is administered in two doses—BNT162b2 (30 µg mRNA) at intervals of 21 days, and mRNA-1273 (100 µg mRNA) at intervals of 28 days—with second doses for both vaccines extendable to 6 weeks.

The third vaccine approved in Europe is the vector vaccine ChAdOx1-nCoV-19 (AZD1222), developed at Oxford University and merchandised by AstraZeneca (trade name Vaxzevria); it was licensed on 29 January 2021, and consists of a replication-deficient chimpanzee adenoviral vector containing the SARS-CoV-2 structural surface glycoprotein antigen (spike protein; nCoV-19) gene [3]. Usually, two doses are administered at intervals of 4–12 weeks, with data from the ChAdOx1 trial suggesting better protection with a longer dosing interval [3]. Overall efficacy using a standard dose of 5 × 10^10^ viral particles after variable intervals was 70.4%, which increased to 90.0% after the first dose was halved [3]. The ChAdOx1-nCoV-19 product information indicates at least 2.5 × 10^8^ infectious units per dose, which makes it difficult to compare current observations with the cited efficacy study. Although approved in Europe for individuals aged ≥ 18 years, recommendations for ChAdOx1-nCoV-19 have undergone changes. Because of initially unclear efficacy at higher ages, the vaccine was primarily recommended for individuals aged < 65 years, while recent observations by Public Health England in March 2021 suggested reasonable effects at ages of 70 years and older as well [4]. Due to a potentially higher risk of thromboembolic events in individuals below the age of 60 years, the German Standing Committee on Vaccination (STIKO) meanwhile recommends the second dose to be administered with an mRNA vaccine in these individuals after first vaccination with ChAdOx1-nCoV-19 (heterologous vaccination scheme).

Here, we show that significant differences exist after different SARS-CoV-2 vaccination schemes with respect to the course and the composition of cellular and serological immune responses, including neutralizing activity against VOCs. Both the strength and the breadth of the immune responses are influenced by the type of vaccine, the use of heterologous vaccination regimes, and the presence or absence of COVID-19-convalescence.

## 2. Materials and Methods

### 2.1. Vaccination and Control Cohorts

The use of blood from healthy human subjects before and after vaccination against SARS-CoV-2, as well as from COVID-19-convalescent subjects, was approved by the Institutional Review Board at Ulm University. For the present study, we compared a reference cohort with a total of 162 COVID-19-convalescent individuals and 8 independent vaccination cohorts including a total of 144 individuals who received various vaccination schemes according to current German guidelines (regarding vaccine selection, choice of vaccine, and vaccination schedule) by the responsible physicians. Only vaccinated individuals without history of diseases or medication affecting systemic immunity were included. For age and gender characteristics of the cohorts, see Table 1.

### 2.2. Serum and PBMC Isolation and Cryopreservation

For serological and neutralization testing, 6 mL of blood from each donor was collected in serum collection tubes (Vacuette, Greiner Bio-One GmbH, Frickenhausen, Germany) after informed consent was given. Serum collection tubes were centrifuged, aliquoted, and cryopreserved at −20 °C until further use. For long-term storage, cryopreservation tubes were transferred to −80 °C. PBMCs were isolated from 10–20 mL heparin blood (Vacutainer Sodium Heparin glass tubes, BD) at specific timepoints via Biocoll density gradient centrifugation.

### 2.3. Enzyme-Linked Immunosorbent Assays

The EUROIMMUN anti-SARS-CoV-2 ELISA assays (EUROIMMUN, Lübeck, Germany) were used for the detection of IgG and IgA against the S1 domain of the SARS-COV-2 spike (S) protein, and IgG against the SARS-CoV-2 nucleocapsid (NCP) protein. OD ratios were calculated based on the sample and calibrator OD values. For all analytes, a ratio < 0.8 was considered to be non-reactive or negative. An OD ratio of ≥1.1 was considered to be positive for all three analytes. Samples with OD ratios > 10 were prediluted in sample buffer at 1:10–1:50 and analyzed again; results were extrapolated accordingly.

### 2.4. Surrogate SARS-CoV-2 Neutralization Test (GenScript)

The principle of this blocking ELISA mimics the virus neutralization process, and qualitatively detects anti-SARS-CoV-2 antibodies, which suppress the interaction between receptor-binding domain (RBD) fragments of the viral spike (S) protein and the angiotensin-converting enzyme 2 (ACE2) protein bound to the surface of a microtiter plate [5]. Samples and controls are preincubated to allow neutralizing antibodies in the serum to bind to either a horseradish peroxidase (HRP)-conjugated wildtype SARS-CoV-2 RBD fragment (HRP-RBD) or to a modified RBD fragment representing the following variants of concern: SARS-CoV-2-spike RBD N501Y (B.1.1.7), SARS-CoV-2-spike RBD E484K, K417N, N501Y (B.1.351), and SARS-CoV-2-spike RBD E484K, K417T, N501Y (P.1). Any unbound HRP-RBD or HRP-RBD bound to non-neutralizing antibodies is captured on a plate. A color reaction, mediated by the substrate TMB, is inversely correlated with the amount of SARS-CoV-2 neutralizing antibodies. Inhibition scores ≥ 30% are considered to be positive. In a modified set of experiments, sample and control sera were directly added to the capture plate without preincubation with HRP-RBD to allow hypothetical autoantibodies to bind to ACE2.

### 2.5. Interferon-Gamma (IFN-γ) Release Assay (IGRA)

The SARS-CoV-2 interferon-gamma (IFN-γ) release assay (IGRA, EUROIMMUN, Lübeck, Germany) was used to detect T-cell-mediated immune response to the SARS-CoV-2 spike antigen in EDTA blood. Results are given in mIU/mL, or as the ratio between OD_CoV-2_IGRA_TUBE_ and OD_CoV-2 IGRA STIM_.

### 2.6. SARS-CoV-2 Peptide Stimulation and FACS Analysis

A total of 0.8–10 × 10^6^ cells/mL were seeded onto a 96-well plate for 2 days in the presence of 100 µL of PepMix SARS-CoV-2 wildtype spike (S-) RBD (JPT, Berlin, Germany), with CEFX Ultra SuperStim Pool (1 µg/mL) as positive or DMSO (2 µL/mL) as negative controls. Purified anti-human CD28 (1 µg/mL, BD, Franklin Lakes, NJ, USA) served as a crosslinker. After incubation, cells were harvested and stained using standard staining procedures. FACS analysis was performed on a BD FACSCelesta (BD, San Jose, CA, USA), and data analysis was carried out using FlowJo software version 10.5.3 (BD, Ashland, OR, USA).

### 2.7. Statistical Analysis

Statistical analysis was performed using Microsoft Excel for Mac, version 16.16.8, and GraphPad Prism version 9.0.0. Summarized data in line graphs are expressed as means ± SEM, or as boxplots with central horizontal lines showing medians, box edges representing interquartile ranges, and whiskers representing the minima and maxima. Dunn’s multiple comparisons test was performed for comparison of more than two types of vaccination at a particular time point. Šídák’s test for multiple comparisons was used for comparison of more than two independent datasets and more than two variables. The Mann–Whitney U test was used for comparison of two datasets. Spearman’s correlation was used to assess associations between neutralization capacity and anti-spike titers.

For more detailed information on cohorts and technical aspects see Appendix A and Methods.

## 3. Results

### 3.1. Temporal Course of Antibody Titers and Neutralization Capacity Significantly Differ between Vaccination Groups and COVID-19-Convalescent Individuals

A total of 144 individual vaccinated individuals from 8 independent vaccination cohorts (A–H) receiving the mRNA vaccines BNT162b2 from BioNTech/Pfizer, mRNA-1273 from Moderna, or the vector vaccine ChAdOx1-nCoV-19 from AstraZeneca were screened for their humoral anti-SARS-CoV-2 immune responses for a time period of up to 15 weeks after their first vaccination (Table 1). A total of 116 vaccinated individuals had no history of COVID-19; 28 were COVID-19-convalescent and analyzed separately. A total of 162 unvaccinated COVID-19-convalescent individuals served as a reference cohort [5,6,7], for which the day of positive pharyngeal swab nucleic acid amplification test (NAT) was considered as timepoint 0. Using identical platforms, we compared the different vaccination cohorts side-by-side, among one another as well as against the reference cohort (cohort I). Most strikingly, between week 6 and up to week 15 after vaccination with the mRNA vaccines, neutralization capacities as well as anti-spike IgG titers ranged significantly higher in the vaccinated individuals than in the convalescent subjects (Figure 1A,B).

Vaccines mRNA-1273 and BNT162b2 showed no significant differences, and were similarly effective with regard to neutralization capacity and anti-spike-IgG, although mRNA-1273 showed a significant advantage over BNT162b2 with regard to anti-spike-IgA titers. For ChAdOx1-nCoV-19-vaccinated individuals, a significantly lower neutralization capacity, as in the reference cohort, could be identified between 3 and 12 weeks after the first vaccination (Figure 1A). In contrast, anti-spike IgG and IgA titers in ChAdOx1-nCoV-19-vaccinated individuals developed almost identically to those in convalescent individuals (Figure 1B,C).

When comparing the individual courses of antibody titers and neutralization capacities, mRNA-vaccinated individuals displayed an impressively uniform and homogeneous pattern compared with the ChAdOx1-nCoV-19-vaccinated individuals (Appendix A).

Within two weeks after the first vaccination, 61% of all BNT162b2-vaccinated individuals had reached a neutralization capacity between 38% and 88% (Appendix A).

Right before the second vaccination (week 3), 95% of the BNT162b2-vaccinated individuals had built up neutralizing capacity, with 33% having reached strong neutralizing capacity > 70% in their serum. One week after their second vaccination (week 4), 81% of vaccinated individuals had built up strong neutralizing capacity > 70%, while 19% had medium neutralizing capacity between 30% and 70%. Five weeks after their first vaccination, 100% of BNT162b2-vaccinated individuals had reached maximum neutralization capacities between 95 and 98%. Similarly, 78% of mRNA-1273-vaccinated individuals had reached a neutralization capacity between 37% and 80% two weeks after their first vaccination (Appendix A).

Right before the second vaccination (week 4), 67% of vaccinated individuals had built up strong neutralizing capacity > 70%, while 33% had medium neutralizing capacity between 30% and 70%. Six weeks after their first vaccination, 100% of mRNA-1273-vaccinated individuals had reached their maximum neutralization capacity between 96 and 98%. In contrast, neutralizing capacity developed at a much slower pace in ChAdOx1-nCoV-19-vaccinated individuals, where three weeks after their first vaccination 40% of all vaccinated individuals exhibited no relevant neutralizing capacity in their serum, and only 12% had built up strong neutralizing capacity > 70% (Appendix A).

Maximum effects were reached at six weeks after the first vaccination, with 54% of all ChAdOx1-nCoV-19-vaccinated individuals having reached medium and 23% having reached strong neutralization capacities between 70 and 79%.

Anti-spike IgG titers reached a first maximum in all vaccinated individuals 3 weeks after their first vaccination (Appendix A).

At this timepoint, anti-spike IgG titers in BNT162b2-vaccinated individuals were four times, and titers in mRNA-1273-vaccinated individuals two times higher than in ChAdOx1-nCoV-19-vaccinated individuals. In BNT162b2- and mRNA-1273-vaccinated individuals, titers exhibited a biphasic course, and reached a second maximum 4 weeks after their second vaccination, when anti-spike IgG titers in BNT162b2-vaccinated individuals were 14 times, and titers in mRNA-1273-vaccinated individuals 26 times higher than in ChAdOx1-nCoV-19-vaccinated individuals. As in convalescent individuals [5,6], we confirmed in vaccinated individuals that anti-spike IgG titers strongly correlated with SARS-CoV-2 neutralization capacity (Appendix A).

Anti-spike IgA titers waned faster in all vaccinated individuals, also reaching two maxima two weeks after the first and second vaccinations in the BNT162b2- and the mRNA-1273-vaccinated individuals, respectively, and 3 weeks after the first vaccination in the ChAdOx1-nCoV-19-vaccinated individuals (Appendix A). Six weeks after first vaccination, IgA titers in BNT162b2-vaccinated individuals were 13 times, and titers in mRNA-1273-vaccinated individuals 43 times higher than in ChAdOx1-nCoV-19-vaccinated individuals.

### 3.2. Previous SARS-CoV-2 Infection Significantly Facilitates Development of a Neutralizing Immune Response after Vaccination with mRNA Vaccines

COVID-19 outbreaks in regional long-term care facilities allowed us to collect serum from both COVID-19-convalescent (cohort C) and COVID-19-naïve individuals (cohort B) in the context of a full vaccination with either BNT162b2 or mRNA-1273. Analysis of anti-NCP IgG titers confirmed their COVID-19 status (Figure 2C and Figure 3C).

Comparing the two cohorts revealed that 3 weeks after their first vaccination, 96.0% of convalescent vaccinated individuals had reached medium and 92% strong neutralization capacity > 70% (Figure 2A). The percentage of COVID-19-naïve vaccinated individuals with a neutralizing capacity > 30% was similar; however, only 33% reached strong neutralization capacity by this timepoint (Figure 2A). Three weeks after their second vaccination (week 6 after first vaccination) this difference disappeared, and strong neutralizing capacity > 70% against wildtype SARS-CoV-2 was detectable in 100% of all vaccinated individuals (Figure 2B).

Although only three COVID-19-convalescent mRNA-1273-vaccinated individuals were available, and did not allow age-matching, basic results were comparable, and confirmed those obtained with BNT162b2-vaccinated individuals (Figure 3A,B).

Most importantly, enhanced neutralizing capacity in BNT162b2-vaccinated convalescent individuals did also cover the three most widespread VOCs in Europe (Figure 2D). Average neutralization capacities in COVID-19-convalescent versus COVID-19-naïve vaccinated individuals were 88.3% versus 75.1% against B.1.17, 73.1% versus 58.5% against B1.351, and 66.8% versus 53.5% against P.1.

### 3.3. Booster Vaccination with an mRNA Vaccine Allows Rapid Development of a Neutralizing Immune Response in ChAdOx1-nCoV-19-Vaccinated individuals

Due to the occurrence of severe immunological adverse events in certain risk groups of ChAdOx1-nCoV-19-vaccinated individuals, health authorities in Germany and other European countries discouraged these groups from receiving a second vaccination with the same vaccine. Instead, they recommended boost-vaccinating these risk groups with an mRNA vaccine. In Germany, the first vaccinated individuals started receiving their booster dose with an mRNA vaccine in the second week of May 2021, generally ~12 weeks after their first vaccination with ChAdOx1-nCoV-19. Right before their second vaccination, >90% of ChAdOx1-nCoV-19-vaccinated individuals exhibited no or medium-level neutralization capacity of <70% (Figure 4A and Appendix A).

One week after booster with either BNT162b2 or mRNA-1273, >85% of vaccinated individuals exhibited strong neutralizing capacity (Figure 4A) and high anti-spike IgG and IgA titers (Figure 4B,C). Both heterologous regimes induced significantly stronger increases in neutralization capacity as well as anti-spike IgG and IgA titer compared with the homologous ChAdOx1/ChAdOx1 regime (Figure 4A–C).

### 3.4. Heterologous Vaccination Regimes Induce Significantly Stronger Neutralizing Capacity against Variants of Concern than Homologous Schemes or Convalescence Alone

Of particular interest is the question of which vaccination regimes may provide the strongest protection against emerging SARS-CoV-2 variants of concern (VOCs). We therefore tested a limited number of individuals from cohorts A, D, F, and G, and the COVID-19-convalescent control cohort I. In general, a gradual loss of neutralizing potency against wildtype SARS-CoV-2- and the VOCs B.1.1.7, B.1.351, and P.1 could be observed (Figure 5A).

As expected, neutralization capacity against all VOCs was significantly lower compared with wildtype neutralization capacity, regardless of the vaccination regime used (Figure 5B–D).

The strongest mean neutralization capacity against VOCs was induced by the heterologous vaccination scheme ChAdOx1/mRNA-1273 (87% against B1.1.7, 85% against B.1.351, and 71% against P.1), followed by ChAdOx1/BNT162b2 (82% against B.1.1.7, 70% against B.1.351 and 55% against P.1). Considering homologous vaccination schemes, mRNA-1273/mRNA-1273 (76% against B1.1.7, 73% against B.1.351 and 56% against P.1) was significantly more effective than BNT162b2/BNT162b2 (63% against B1.1.7, 59% against B.1.351 and 51% against P.1) and ChAdOx1/ChAdOx1 (48% against B1.1.7, 57% against B.1.351 and 15% against P.1) at neutralizing VOCs in vitro. Importantly, compared with individuals vaccinated with a heterologous regime, unvaccinated COVID-19-convalescent individuals exhibited significantly weaker neutralization capacities against B.1.1.7 (67%) and P.1 (35%), whereas neutralization capacity against B.351 (75%) did not significantly differ from that in vaccinated individuals (Figure 5A–D).

### 3.5. SARS-CoV-2-Specific T-Cell Responses Significantly Differ between Vaccination Groups

In addition to serological endpoints, we also tested some aspects of the cellular immune response in the various vaccination groups. Considering the time between first vaccination and analysis, there was a clear trend to increasing SARS-CoV-2-spike-specific T-cell responses in both mRNA vaccination groups (Figure 6A,B).

Importantly, the three convalescent mRNA-1273-vaccinated individuals described above also showed stronger T-cell responses compared with COVID-19-naïve vaccinated individuals (Figure 6B). In addition, we included one case of an accidental “inverse” heterologous vaccination scheme consisting of a first vaccination with BNT162b2 and, three weeks later, a second dose of ChAdOx1 (Figure 6A). Although this case has anecdotal character only, the T-cell response was comparable with that in homologous BNT162b2/BNT262b2-vaccinated individuals. Figure 6C summarizes individual T-cell responses in the four ChAdOx1-primed cohorts E, F, G, and H.

Peak IFN-γ secretion was significantly stronger in individuals receiving a heterologous (ChAdOx1/BNT162b2 and ChAdOx1/mRNA-1273) compared with individuals receiving a homologous (ChAdOx1/ChAdOx1, BNT162b2/BNT162b2 and mRNA-1273/mRNA-1273) vaccination regime (Figure 6D).

FACS-based analysis confirmed that T-cell activation after ChAdOx1-nCoV-19-based immunizations involved both CD4^+^ T helper and CD8^+^ cytotoxic T cells (Appendix A).

Expression of the activation marker CD69 was consistently upregulated in both T-cell populations after stimulation with SARS-CoV-2 spike peptides in vaccinated individuals having received a ChAdOx1-based immunization (Appendix A), but not in unvaccinated control individuals (Appendix A).

## 4. Discussion

To prevent spread of SARS-CoV-2, many countries have taken drastic measures by starting vast vaccination programs utilizing various approved SARS-CoV-2 vaccines. Since it may be politically tempting to convey all vaccines as equally effective, it is important to provide an independent view on both adverse effects and real-life efficacy by monitoring and comparing immune responses to various vaccines, particularly in the face of emerging virus mutants [8]. For the eventual control of the current SARS-CoV-2 pandemic, it is of paramount interest to not only know how rapidly and efficiently vaccinated individuals build up immune responses, but also how long they keep a sufficiently strong and neutralizing immunity against SARS-CoV-2 and the emerging variants of concern (VOCs). Although the current study included real-life cohorts with small numbers of vaccinated individuals, so that a potential impact of host factors—such as HLA haplotype or previous diseases—cannot be absolutely excluded, it may help to answer the first of these two important questions.

One major finding of our study was that within three weeks after their first vaccination, and before receipt of booster immunizations, vaccinated individuals who had received an mRNA vaccine displayed a rapid and relatively uniform increase not only in terms of SARS-CoV-2 neutralization potential, but also regarding anti-spike antibody titers. Three weeks after their first vaccination, IgG and IgA titers in mRNA vaccinated individuals were already 2–4 times higher than in ChAdOx1-nCoV-19-vaccinated individuals. Multivariate analyses among indicated timepoints revealed a highly significant superiority of both mRNA vaccination groups over not only the reference cohort of COVID-19-convalescent individuals, but also the vector vaccine ChAdOx1-nCoV-19. Within three weeks after their first vaccinations, neutralization capacity in the ChAdOx1-nCoV-19 cohort appeared very heterogeneous compared with the mRNA vaccine cohorts, and ranged significantly below that of convalescent individuals. Although anti-spike antibody titers started to decrease three weeks after second vaccination, with a relatively short half-life of 1–2 weeks, up to the current observation period of 15 weeks after their first vaccination, neutralization capacities of BNT162b2-vaccinated individuals remained at a high level [6,9,10]. A possible explanation for the more stable kinetics of neutralization capacity in mRNA-vaccinated individuals compared with antibody titers may be a progressive affinity maturation of antibodies specific to the receptor-binding domain (RBD) of SARS-CoV-2 [11]. As an alternative explanation, the occurrence of autoantibodies against angiotensin-converting enzyme 2 (ACE2)—the molecule allowing SARS-CoV-2 cell entry [12]—were discussed. Such autoantibodies have been demonstrated in severely ill COVID-19 patients [13]. However, although in theory they may also evolve after vaccination, this possibility was experimentally excluded.

Our data further demonstrate that the mean neutralization capacity after vaccination with mRNA vaccines is significantly higher than the mean neutralization capacity in convalescent plasma donors observed over several months after positive pharyngeal swab [5,6]. Initial hints that the neutralizing capacity may be equal or even stronger in vaccinated than in convalescent individuals were already assumed during the phase 1 trial of BNT162B2 [14]. Our study reveals that a strong neutralization capacity after vaccination with mRNA vaccines is achieved much more rapidly, and that this effect is particularly strong in COVID-19-convalescent individuals. Importantly, our study extends the results from recent papers describing neutralizing antibody responses after SARS-CoV-2 infection in general as well as cross-variant neutralizing antibody responses against B.1.351 elicited by mRNA vaccination boosts after SARS-CoV-2 infection [15]. We demonstrate that BNT162b2 vaccination induces a significantly stronger neutralization capacity in COVID-19-convalescent compared with COVID-19-naïve individuals, and that it covers the three most widespread variants of concern in Europe: B.1.1.7, B.1.351, and P.1. On the one hand, our results show how important vaccinations are in convalescent individuals as well; on the other hand, our data suggest that a single booster vaccination using an mRNA vaccine may be an effective and practicable way for rapid enhancement of neutralizing antibody titers in convalescent plasma donors. The use of convalescent plasma is still considered a potentially important pillar for the therapy of moderately-to-severely ill COVID-19 patients, although its benefit appears to be closely linked to sufficiently high antibody titers [16,17,18]. Recently, the FDA updated their recommendations regarding the emergency use of high-titer COVID-19 convalescent plasma, in which neutralization capacities > 68% with the neutralization assay used in the current study were considered sufficiently strong to induce beneficial effects in COVID-19 patients (https://www.fda.gov/media/141477/download (accessed on 1 July 2021)).

As mentioned initially, thromboembolic adverse events after vaccination with ChAdOx1-nCoV-19 [19] have led to the recommendation to use a heterologous prime-boost vaccination strategy, recently proven efficient in animal models [20] and currently being examined in a COVID-19 vaccine alternating-dose study in the UK (https://www.nihr.ac.uk/news/worlds-first-covid-19-vaccine-alternating-dose-study-launches-in-uk/26773 (accessed on 1 July 2021)). Initial interim analyses reveal that heterologous vaccination schedules can induce greater systemic reactogenicity following the booster dose than their homologous counterparts [21,22], but particularly comparative data on the immunogenicity of different regimes have not been published so far. In our study, we acquired data on some of the first ChAdOx1-nCoV-19-vaccinated individuals who received an mRNA booster dose due to a change in vaccination recommendations in Germany. Considering both humoral and cellular immune responses, our data confirm the suggested efficacy of this heterologous vaccination approach. In fact, the dynamics of antibody and neutralization titers were reminiscent of those of convalescent individuals receiving a single dose of an mRNA vaccine. Particularly, neutralizing capacities against the most widespread variants of concern—B.1.1.7, B.1.351, and P.1—were significantly enhanced by heterologous compared with homologous vaccination regimes. Similar results were obtained when considering the cellular immune response. Average SARS-CoV-2-specific peak IFN-γ responses were significantly stronger in vaccinated individuals receiving a heterologous vaccination scheme compared with vaccinated individuals receiving homologous vaccination regimes. Particularly, homologous vaccination with ChAdOx1-nCoV-19 resulted in significantly lower peak IFN-γ responses than heterologous vaccination approaches. A recent study demonstrated that a single dose of ChAdOx1-nCoV-19 is able to induce a robust Th1-based CD4^+^ T-cell response [23]. Our data demonstrate that one dose of an mRNA vaccine is sufficient to reactivate both CD4^+^ and CD8^+^ memory T-cell responses against SARS-CoV-2 in ChAdOx1-nCoV-19-vaccinated individuals within one week, similarly to what has been previously described for convalescent individuals [24].

In summary, the current study is among the first performing a simultaneous side-by-side comparison of BNT162b2, mRNA-1273, and ChAdOx1-nCoV-19, both in homologous and heterologous settings, regarding their immunogenicity in COVID-19-naïve and -convalescent individuals. We found that humoral and cellular immune responses evolved significantly faster and involved higher neutralizing antibody titers against SARS-CoV-2—including the most widespread VOCs—with both mRNA vaccines compared to ChAdOx1-nCoV-19. Similarly, the ChAdOx1-nCoV-19 regime resulted in neutralization capacities that ranged even below those of COVID-19-convalescent individuals, whereas neutralization capacities with mRNA vaccine regimes reached high and stable plateaus > 95% within 3–4 weeks after the first vaccination. Importantly, both COVID-19-convalescent individuals and individuals having received a first vaccination with ChAdOx1-nCoV-19 reached strong neutralization capacity within one week of a single mRNA vaccine booster. Our results suggest that both COVID-19-convalescent individuals and ChAdOx1-nCoV-19-primed individuals may acquire strong neutralizing immunity against SARS-CoV-2 and VOCs after a single mRNA vaccine booster. Importantly, this approach may also represent an effective way to rapidly qualify or requalify convalescent plasma donors with high titers of broadly neutralizing antibodies.

## Figures and Tables

**Figure 1 vaccines-09-00918-f001:**
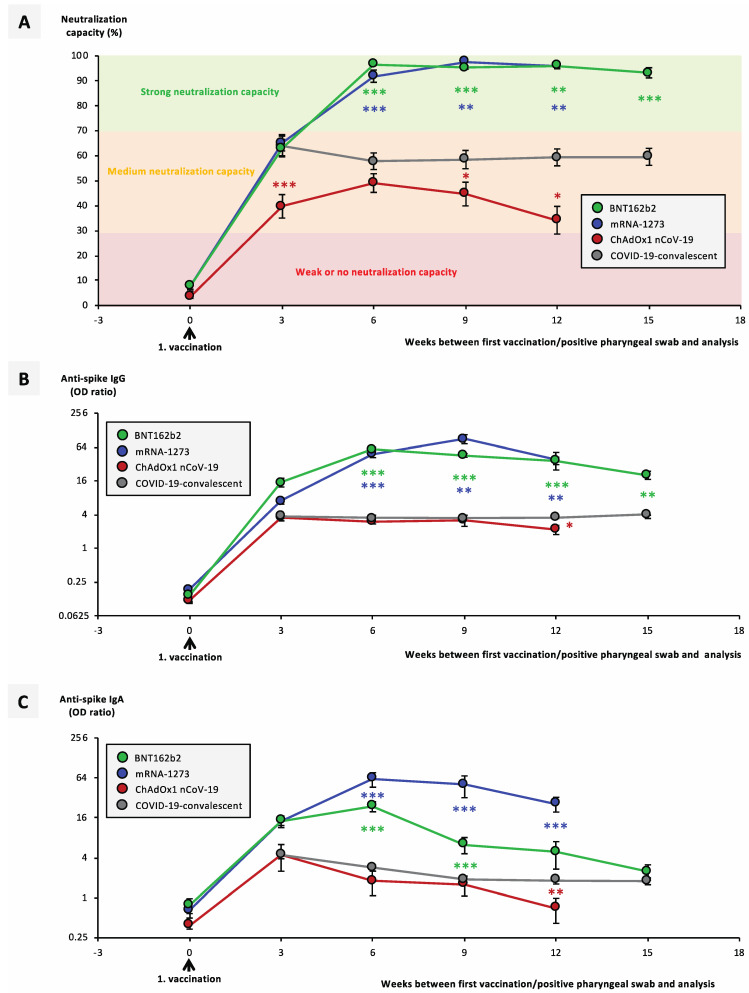
Average antibody and neutralization titers after different anti-SARS-CoV-2 vaccination regimes. Serum samples from 21 BNT162b2- (cohort A), 10 mRNA-1273- (cohort D), and 29 ChAdOx1-nCoV-19 (cohort E)-vaccinated individuals were collected at different timepoints after their first vaccination, as indicated. Second vaccination was given 3 weeks after first vaccination in BNT162b2-, and 4 weeks after first vaccination in mRNA-1273-vaccinated individuals. Serum samples from 162 COVID-19-convalescent individuals, collected at different timepoints after positive pharyngeal swab, served as the control cohort (cohort I). Plotted are average results for (**A**) neutralization of ACE2–RBD interaction, (**B**) anti-spike IgG titers, and (**C**) anti-spike IgA titers. (**A**) Neutralization capacities > 30% were considered positive, neutralization capacities > 70% were considered strong. Error bars indicate SEM; significance levels shown are between vaccinated individuals (colored circles) and convalescent individuals (grey circles) only. Statistical analysis at each timepoint was performed using Dunn’s multiple comparisons test. Significance levels between ChAdOx1-nCoV-19-vaccinated individuals and BNT162b2- or mRNA-1273-vaccinated individuals ranged at the same level as those between convalescent individuals and BNT162b2- or mRNA-1273-vaccinated individuals. Significance levels were * *p* < 0.05, ** *p* < 0.005, and *** *p* < 0.0005. Abbreviations: COVID-19: coronavirus disease 2019; ACE2: angiotensin-converting enzyme 2; RBD: receptor-binding domain.

**Figure 2 vaccines-09-00918-f002:**
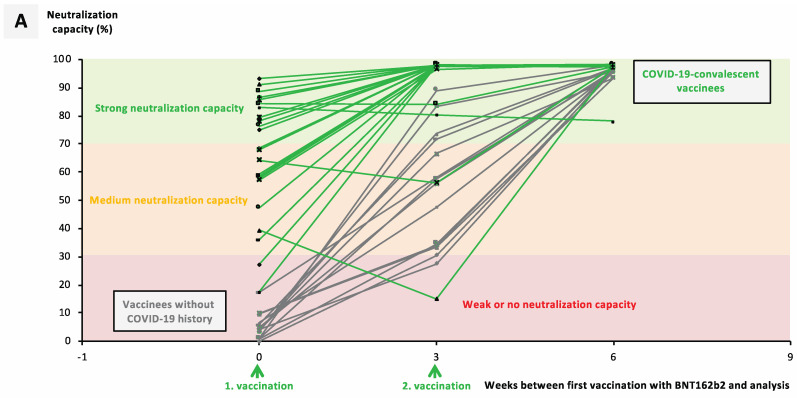
Impact of previous SARS-CoV-2 infection on the neutralizing capacity after vaccination with BNT162b2. Serum samples from 15 BNT162b2-vaccinated individuals without history of COVID-19 (cohort B) and 25 COVID-19-reconvalescent BNT162b2-vaccinated individuals (cohort C) were collected at different timepoints after their first vaccination, as indicated. (**A**) Line graphs show individual courses for the neutralization capacity of ACE2–RBD interaction. (**B**) Boxplots show median neutralization capacities at the time of first vaccination (0 weeks), second vaccination (3 weeks), and 3 weeks after second vaccination (6 weeks). (**C**) Boxplots show median anti-NCP IgG titers for COVID-19-naïve versus COVID-19-convalescent donors. (**D**) Boxplots show neutralization capacities against RBD variants as indicated in BNT162b2-vaccinated COVID-19-naïve versus COVID-19-convalescent individuals. Box central horizontal lines in (**B**,**C**) indicate medians, box borders represent IQR, and whiskers indicate minima and maxima. For statistics, Šídák’s test for multiple comparisons (**B**,**D**), and the Mann–Whitney U test (**C**) were used. Significance levels were ** *p* < 0.005, *** *p* < 0.0005, and **** *p* < 0.00005. Abbreviations: COVID-19: coronavirus disease 2019; ACE2: angiotensin-converting enzyme 2; RBD: receptor-binding domain; IQR: interquartile range; n.s.: not significant.

**Figure 3 vaccines-09-00918-f003:**
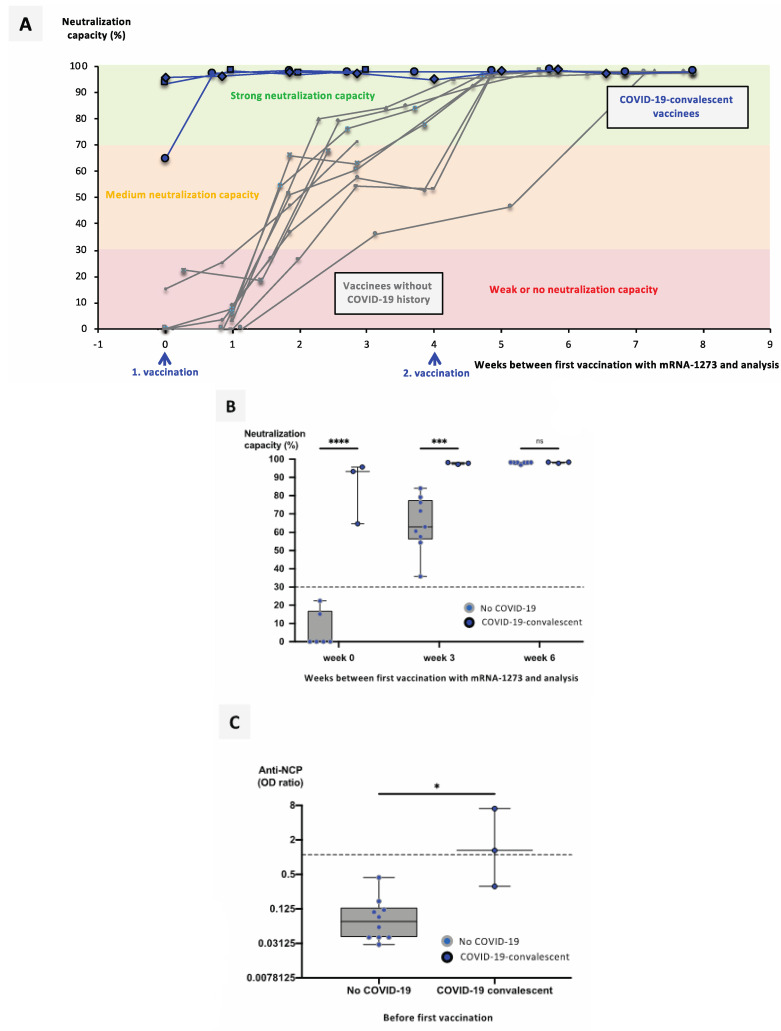
Impact of a previous SARS-CoV-2 infection on the neutralizing immune response to vaccination with mRNA-1273. Serum samples from 10 mRNA-1273-vaccinated individuals without history of COVID-19 and 3 COVID-19-reconvalescent mRNA-1273-vaccinated individuals were collected at different timepoints after their first vaccination, as indicated. (**A**) Line graphs show individual courses for the neutralization capacity of ACE2–RBD interaction. (**B**) Boxplots show median neutralization capacities at the time of first vaccination (0 weeks), as well as 3 weeks and 6 weeks after first vaccination. (**C**) Boxplots show median anti-NCP IgG titers for COVID-19-negative versus COVID-19-convalescent donors. Box central horizontal lines indicate medians, box borders represent IQR, and whiskers indicate minima and maxima. For statistics, Šídák’s test for multiple comparisons (**B**) and the Mann–Whitney U test (**C**) were used. Significance levels were * *p* < 0.05, *** *p* < 0.0005, and **** *p* < 0.00005. Abbreviations: COVID-19: coronavirus disease 2019; ACE2: angiotensin-converting enzyme 2; RBD: receptor-binding domain; IQR: interquartile range; n.s.: not significant.

**Figure 4 vaccines-09-00918-f004:**
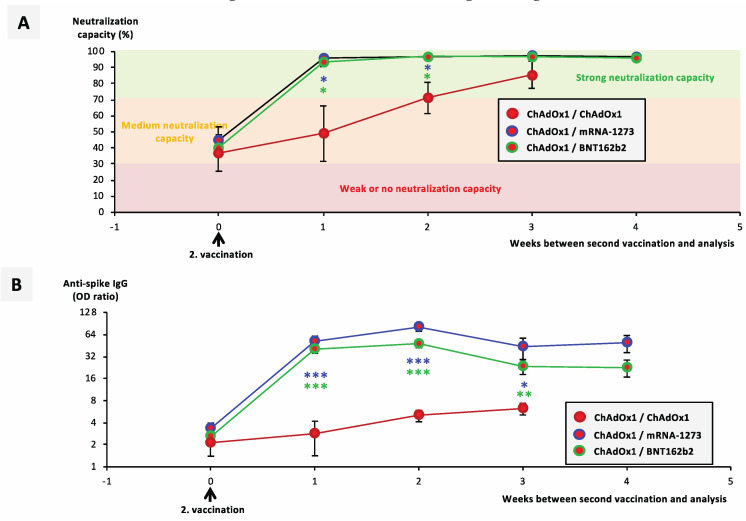
Impact of heterologous vaccination with ChAdOx1-nCoV-19 and an mRNA vaccine on neutralizing immune responses. Serum samples from 5 vaccinated individuals having received a homologous ChAdOx1-nCoV-19 vaccination (cohort H) and 36 vaccinated individuals having received a heterologous vaccination scheme (cohorts F and G) were collected at different timepoints after their second vaccinations. Line graphs show average courses for (**A**) the neutralization capacity of ACE2–RBD interaction, (**B**) anti-spike IgG titers, and (**C**) anti-spike IgA titers from 26 ChAdOx1/BNT162b2, 10 ChAdOx1/mRNA-1273, and 5 ChAdOx1/ChAdOx1 vaccinated individuals at different timepoints after their second vaccinations, as indicated. Statistical analysis at each timepoint was performed using Dunn’s multiple comparisons test. Significance levels were * *p* < 0.05, ** *p* < 0.005, and *** *p* < 0.0005. Abbreviations: ACE2: angiotensin-converting enzyme 2; RBD: receptor-binding domain.

**Figure 5 vaccines-09-00918-f005:**
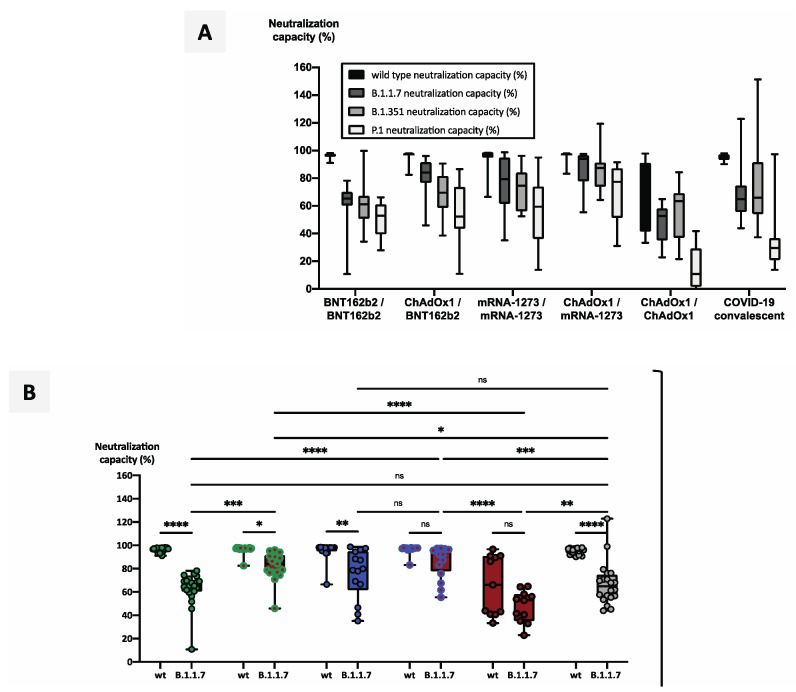
Impact of vaccination regimes on the neutralizing capacity against variants of concern. Serum samples from 23 subjects after homologous vaccination with BNT162b2, 23 subjects after heterologous vaccination with ChAdOx1/BNT162b2, 23 subjects after homologous vaccination with mRNA-1273, 23 subjects after heterologous vaccination with ChAdOx1/mRNA-1273, and 11 subjects after homologous vaccination with ChAdOx1 were collected 2 weeks after their second vaccinations. Serum samples from 20 unvaccinated COVID-19-convalescent individuals, collected at a median time of 14 weeks after diagnosis, served as control cohort. Samples were analyzed for neutralization capacity against wildtype RBD and three RBD variants of concern. Boxplots in panel (**A**) summarize neutralization capacities in the five vaccination cohorts and in an unvaccinated COVID-19-convalescent cohort. (**B**–**D**) Boxplots show neutralization capacities against wildtype RBD and (**B**) the B.1.1.7 RBD variant, (**C**) the B.1.351 RBD variant, and (**D**) the P.1 RBD variant. Box central horizontal lines indicate medians, box borders represent IQR, and whiskers represent minima and maxima. Significance levels were tested by Šídák’s test for multiple comparisons, with * *p* < 0.05, ** *p* < 0.005, *** *p* < 0.0005, and **** *p* < 0.00005. Abbreviations: IQR: interquartile range; ns: not significant; RBD: receptor-binding domain.

**Figure 6 vaccines-09-00918-f006:**
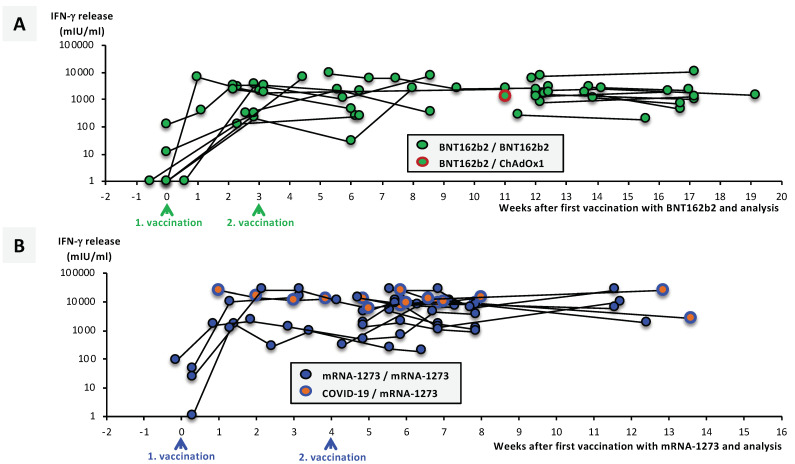
Comparison of SARS-CoV-2-specific T-cell response after homologous and heterologous vaccination. A total of 336 heparin blood samples from (**A**) 24 BNT162b2-vaccinated individuals, (**B**) 15 mRNA-1273-vaccinated individuals, and (**C**) 41 ChAdOx1-nCoV-19-primed vaccinated individuals were collected at different time points after their first vaccination, as indicated, and incubated overnight with a SARS-CoV-2-spike peptide mix as described in the Materials and Methods section. Then, supernatants were harvested and IFN-γ concentrations measured via ELISA. (**A**–**C**) Dot plots show IFN-γ concentrations at individual timepoints, as indicated. Lines connect different timepoints for corresponding individuals. Panel (**A**) also shows one case of an inverse heterologous vaccination with BNT162b2 and ChAdOx1-nCoV-19; in panel (**B**), 3 cases of COVID-19-convalescent mRNA-1273-vaccinated individuals are shown. Panel (**C**) shows results from a total of 205 timepoints from 41 individual ChAdOx1-nCoV-19-primed vaccinated individuals having continued with either homologous or heterologous vaccination regimes, as indicated by different color codes. Black lines indicate IFN-γ courses after first, and colored lines after second vaccinations. (**D**) Boxplots show peak IFN-γ concentrations 1–2 weeks after second vaccination in the various vaccination groups, as indicated. Box central horizontal lines indicate medians, box borders indicate IQR, and whiskers represent minima and maxima. Significance levels were determined by Dunn’s multiple comparisons test, with * *p* < 0.05, ** *p* < 0.005, and **** *p* < 0.00005. Abbreviations: IFN-γ: interferon-gamma; IQR: interquartile range; ns: not significant.

**Table 1 vaccines-09-00918-t001:** Age and gender characteristics of different vaccination cohorts and a reference cohort of unvaccinated COVID-19-convalescent individuals. For comparison of serological immune responses after first vaccination, cohorts A, D, and E were compared with cohort I. For comparison of serological immune responses in COVID-19-naive and COVID-19-convalescent vaccinated individuals, cohort B was compared with cohort C. For comparison of serological immune responses after second vaccination, cohorts A, D, and H were used to represent homologous vaccination schemes, and cohorts F and G to represent heterologous vaccination schemes.

Cohort Code	A	B	C	D	E	F	G	H	I
Vaccine Type/ COVID-19 Convalescence	BNT162b2/ BNT162b2	BNT162b2/ BNT162b2 COVID-19-Naive	BNT162b2/ BNT162b2 COVID-19-Convalescent	mRNA-1273/ mRNA-1273	ChAdOx1/-	ChAdOx1/ BNT162b2	ChAdOx2/ mRNA-1273	ChAdOx1/ ChAdOx1	Unvaccinated Reference Cohort COVID-19-Convalescent
Number of Individuals	21	15	25	13	29	26	10	5	162
Number Females (%)	17 (81.0%)	12 (80.0%)	21 (84.0%)	8 (61.5%)	25 (86.2%)	2 (7.7%)	7 (70.0%)	3 (60.0%)	76 (46.9%)
Number Males (%)	4 (19.0%)	3 (20.0%)	4 (16.0%)	5 (38.5%)	4 (13.8%)	24 (92.3%)	3 (30.0%)	2 (40.0%)	86 (53.1%)
Average Age (Range)	45 (26*–*65)	47 (26*–*64)	46 (21*–*73)	51 (34*–*61)	45 (24*–*64)	44 (22*–*64)	33 (21*–*47)	53 (46*–*63)	32 (19*–*61)

## Data Availability

Data may be available upon request.

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
