# Peer review of "mRNA Vaccines Enhance Neutralizing Immunity against SARS-CoV-2 Variants in Convalescent and ChAdOx1-Primed Subjects"

_vaccines, 2021, doi:10.3390/vaccines9080918_

Round 1
Reviewer 1 Report
Title of the manuscript: mRNA vaccines enhance neutralizing immunity against SARS-CoV-2 variants in convalescent and ChAdOx1-primed subject
Manuscript ID: vaccines-1314960
Evaluation Summary: The present work by Fabricius D et al., is aimed to study the efficacy of various vaccinations regimes by running side by side comparisons of immunological responses of around 144 individuals with the mRNA vaccines BNT162b2 or mRNA-1273 and the vector vaccine ChAdOx1-nCoV-19, either alone or in combination or in the context of COVID-19-convalescence. Data suggests that neutralizing capacity against VOCs were significantly stronger with mRNA vaccines as compared with COVID-19-convalescent individuals or vaccinees receiving the vector vaccine ChAdOx1 nCoV-19 and that Booster immunizations with mRNA vaccines triggered strong and broadly neutralizing 27 antibody and IFN-g responses in 100% of vaccinees. However, authors has noted that heterologous vaccination regimes and convalescent booster regimes using mRNA vaccines allow enhanced protection against VOC’s which is important at this stage.
Overall, the manuscript has is well written with a goal to search for proportion of individuals with SARS-CoV-2 neutralizing antibodies after various vaccinations regimes. Authors have summarized results of study well and analyzed neutralizing antibody titers against SARS-CoV-2 wt vs VOCs. Conclusions drawn from the analysis were clear. But I, have few concerns outlined below to be addressed by the authors:
Strengths of the study:
- Assessment of efficacy of heterologous vaccination regimes. Given the current situation of vaccines shortage it is important to know if heterologous vaccination would generate protective responses.
- And if heterologous vaccination would generate protective responses, are these protective responses neutralize variants of concern (VOCs).
- Assessment of vaccination requirement in previously SARS-CoV-2 infected individuals or convalescent individuals.
Weakness: Not using appropriate tools to assess neutralizing capacity
Recommendations/Comments to authors:
Major:
- SARS-CoV-2 Neutralization Antibody Detection Kit used by authors is just a surrogate virus neutralization test (sVNT), a serological assay to determine the presence of RBD blocking antibodies that compete for human ACE2 binding but does not assess actual virus neutralization capacity. There are several manuscripts presenting non-RBD targeting antibodies possessing neutralizing capacity (please see below for references). So, I recommend authors to include these references and discuss the same in discussion as one of the limitations of the assay or study.
- Suryadevara, Naveenchandra, Swathi Shrihari, Pavlo Gilchuk, Laura A. VanBlargan, Elad Binshtein, Seth J. Zost, Rachel S. Nargi et al. "Neutralizing and protective human monoclonal antibodies recognizing the N-terminal domain of the SARS-CoV-2 spike protein." Cell (2021).
- Chi, Xiangyang, Renhong Yan, Jun Zhang, Guanying Zhang, Yuanyuan Zhang, Meng Hao, Zhe Zhang et al. "A neutralizing human antibody binds to the N-terminal domain of the Spike protein of SARS-CoV-2." Science 369, no. 6504 (2020): 650-655.
- Usually, sVNT demonstrated a high non-neutralizing antibody detection rate. This has been evaluated in case SARS-2 sVNT kits as well and found that agreement between sVNT and PRNT-50 was moderate. Hence, conclusions about neutralization capacity are contradictory by using sVNT. Can authors comment on this?
- I recommend authors to perform at least pseudo neutralization assay for few samples from each group and correlate that with sVNT assay results to check for consistency of results which will enhance the quality of the manuscript.
Minor comments:
Results:
- Figure 3B and C legends does not match with the figure colors
Discussion:
- Line 366 remove it to be first study as there are several reports comparing vaccine responses side by side.
Author Response
Reviewer #1: (Comment & Reponse)
Major:
- SARS-CoV-2 Neutralization Antibody Detection Kit used by authors is just a surrogate virus neutralization test (sVNT), a serological assay to determine the presence of RBD blocking antibodies that compete for human ACE2 binding but does not assess actual virus neutralization capacity. There are several manuscripts presenting non-RBD targeting antibodies possessing neutralizing capacity (please see below for references). So, I recommend authors to include these references and discuss the same in discussion as one of the limitations of the assay or study.
- Suryadevara, Naveenchandra, Swathi Shrihari, Pavlo Gilchuk, Laura A. VanBlargan, Elad Binshtein, Seth J. Zost, Rachel S. Nargi et al. "Neutralizing and protective human monoclonal antibodies recognizing the N-terminal domain of the SARS-CoV-2 spike protein." Cell (2021).
- Chi, Xiangyang, Renhong Yan, Jun Zhang, Guanying Zhang, Yuanyuan Zhang, Meng Hao, Zhe Zhang et al. "A neutralizing human antibody binds to the N-terminal domain of the Spike protein of SARS-CoV-2." Science 369, no. 6504 (2020): 650-655.
First, we thank reviewer #1 for the extensive evaluation of our manuscript. We are aware that due to the high number of subjects included in our current study, we used a surrogate neutralization test only. However, as pointed out in our manuscript, this surrogate neutralization test platform is meanwhile recommended by the FDA to characterize and identify reconvalescent plasma donors with high neutralization titers. In addition, earlier this year we have performed a side-by-side validation of this platform using both a wildtype virus plaque neutralization test platform (using African green monkey VeroE6 cells) and a pseudovirus test platform (using human Caco-2 cells) (Jahrsdörfer et al., Journal of Immunology, 2021). This validation clearly demonstrated a strong correlation between the surrogate neutralization test platform used in the current study and the wildtype virus platform, which is widely considered as the gold standard for defining SARS-CoV-2 neutralization potency. We therefore consider our results as valid and reliable and do not feel additional methods are required to confirm them.
Nevertheless, we agree that this fact may be pointed out our more clearly and have therefore modified the respective passages in our manuscript and included the two references mentioned by reviewer #1.
- Usually, sVNT demonstrated a high non-neutralizing antibody detection rate. This has been evaluated in case SARS-2 sVNT kits as well and found that agreement between sVNT and PRNT-50 was moderate. Hence, conclusions about neutralization capacity are contradictory by using sVNT. Can authors comment on this?
As described above, in our hands the correlation between the surrogate neutralization test used in our study and the gold standard “plaque reduction neutralization test” (PRNT50) was strong and highly significant (Jahrsdörfer et al., Journal of Immunology, 2021). However, as with other in vitro-test systems caution should always be exercised when using immunological results to predict how individuals or individual cells may respond to infection with a wildtype virus. A series of factors may impact on an individual immune response in the “real life setting” including the HLA type of cells or individuals, a previous history of diseases or medication and many others. We therefore included this caveat into the discussion section of our manuscript.
- I recommend authors to perform at least pseudo neutralization assay for few samples from each group and correlate that with sVNT assay results to check for consistency of results which will enhance the quality of the manuscript.
We do not recognize how the inclusion of a few samples would add to the current manuscript and study. After all, the point of our recent validation of the surrogate neutralization test used in the current study (Jahrsdörfer et al., Journal of Immunology, 2021) was to get rid of laborious and time-consuming additional test platforms.
Minor comments:
Results:
- Figure 3B and C legends does not match with the figure colors
This error has been corrected, a modified figure 3 has been created and inserted into the manuscript file.
Discussion:
- Line 366 remove it to be first study as there are several reports comparing vaccine responses side by side.
We agree and have modified the wording accordingly in the manuscript file.
Reviewer 2 Report
In this very interesting and original study, the authors identified a group of individuals (i.e., COVID-19-convalescent individuals and individuals vaccinated with a 1st vaccination with ChAdOx1-nCoV-19 receiving a single mRNA vaccine booster) potentially able to donate convalescent plasma with high levels of neutralizing antibodies against SARS-CoV-2 VOC.
This study is particularly topical, considering the current new COVID-19 wave and the need of collecting rapidly high-titer convalescent plasma against VOC.
I have no particular comments.
Author Response
Reviewer #2: (Comment & Response)
In this very interesting and original study, the authors identified a group of individuals (i.e., COVID-19-convalescent individuals and individuals vaccinated with a 1st vaccination with ChAdOx1-nCoV-19 receiving a single mRNA vaccine booster) potentially able to donate convalescent plasma with high levels of neutralizing antibodies against SARS-CoV-2 VOC.
This study is particularly topical, considering the current new COVID-19 wave and the need of collecting rapidly high-titer convalescent plasma against VOC.
I have no particular comments.
We thank reviewer #2 for the positive evaluation.
Reviewer 3 Report
In this study, Fabricius et al present a very interesting finding that heterologous immunization with ChAdOx1-prime followed by boost with mRNA vaccine shows the most promising neutralization titers as well as cellular immune response, which particularly impressive against the variants of concern. This study has broad implications in terms of ongoing vaccinations. The paper is well-written and is easy to follow. I recommend that the paper be published as such.
I have a minor comment, which I believe will add an important missing point in the paper: the authors show that COVID-19 convalescent subjects trigger very strong immune response upon administration of only one dose of the mRNA vaccine. I am curious to know how such convalescent subjects respond to one dose of the ChAdOx1 vaccine, considering that one dose of the same in naïve individuals only elicits poor response. This data is particularly helpful for countries like India where a significant fraction of the population got infected by COVID in the devastating second wave and have subsequently been administered ChAdOx1 vaccine. Of course if the authors do not have access to such subjects, then this can be skipped.
Author Response
Reviewer #3: (Comments & Responses)
In this study, Fabricius et al present a very interesting finding that heterologous immunization with ChAdOx1-prime followed by boost with mRNA vaccine shows the most promising neutralization titers as well as cellular immune response, which particularly impressive against the variants of concern. This study has broad implications in terms of ongoing vaccinations. The paper is well-written and is easy to follow. I recommend that the paper be published as such.
We thank reviewer #3 for the positive evaluation.
I have a minor comment, which I believe will add an important missing point in the paper: the authors show that COVID-19 convalescent subjects trigger very strong immune response upon administration of only one dose of the mRNA vaccine. I am curious to know how such convalescent subjects respond to one dose of the ChAdOx1 vaccine, considering that one dose of the same in naïve individuals only elicits poor response. This data is particularly helpful for countries like India where a significant fraction of the population got infected by COVID in the devastating second wave and have subsequently been administered ChAdOx1 vaccine. Of course if the authors do not have access to such subjects, then this can be skipped.
We agree with reviewer #3 that inclusion of COVID-19-convalescent subjects being vaccinated with ChAdOx1 would definitely add interesting information to the current study. Unfortunately, as expected by reviewer #3, we do not currently have access to such subjects. Nevertheless, we now included into the revised manuscript an additional cohort of subjects having received full vaccination with ChAdOx1 (two vaccinations) and compared their neutralization capacity against SARS-CoV-2 including variants of concern with the other homologous and heterologous vaccination cohorts. To this purpose a new figure 5 was created and inserted into the manuscript file.